# Learning Distributionally Robust Tractable Probabilistic Models in Continuous Domains

**Hailiang Dong**[1]        **James Amato**[1]        **Vibhav Gogate**[1]        **Nicholas Ruozzi**[1]

[1]Computer Science Dept., The University of Texas at Dallas, Texas, 75080, USA

## Abstract

Tractable probabilistic models (TPMs) have attracted substantial research interest in recent years, particularly because of their ability to answer various reasoning queries in polynomial time. In this study, we focus on the distributionally robust learning of continuous TPMs and address the challenge of distribution shift at test time by tackling the adversarial risk minimization problem within the framework of distributionally robust learning. Specifically, we demonstrate that the adversarial risk minimization problem can be efficiently addressed when the model permits exact log-likelihood evaluation and efficient learning on weighted data. Our experimental results on several real-world datasets show that our approach achieves significantly higher log-likelihoods on adversarial test sets. Remarkably, we note that the model learned via distributionally robust learning can achieve higher average log-likelihood on the initial uncorrupted test set at times.

## 1 INTRODUCTION

Tractable Probabilistic Models (TPMs), including Probabilistic Sentential Decision Diagrams (PSDDs) (Kisa et al., 2014), Arithmetic Circuits (ACs) (Darwiche, 2003), Sum-Product Networks (SPNs) (Poon and Domingos, 2011), and Cutset Networks (CNs) (Rahman et al., 2014), have gained significant attention and research interest in recent years. These models fall under a unified framework known as probabilistic circuits (Choi et al., 2020), and they offer promising solutions for modelling the uncertainties. One of the key features that makes these models particularly attractive is their ability to perform certain inferences in polynomial time such as exact likelihood calculations, or in some cases, finding the most probable assignment for unobserved variables given evidence (Rahman et al., 2019; Dong et al., 2023; Molina et al., 2018).

The learning of TPMs mainly relies on the Maximum Likelihood Estimation (MLE) framework, which assumes the training data is free of corruption and noise, while effectively representing the underlying data distribution. However, this assumption can fail in practice due to a wide range of factors such as measurement errors, label noise and sample selection bias. Existing research on robust learning of TPMs has mostly been limited to binary or discrete domains (Peddi et al., 2022). In this work, we address the challenges of learning robust TPMs in continuous domains.

The Robust MLE framework provides a foundation for exploring the robust learning of TPMs (Bertsimas and Nohadani, 2019), which can be further divided into Adversarial Robust MLE (ARM) and Distributionally Robust MLE (DRM). ARM optimizes against the worst-case scenarios among neighboring data observations within a certain distance, treating each neighboring point with equal importance, a practice that might not align with real-world scenarios. On the other hand, DRM faces an inherent challenge where the inner minimization problem is often intractable. Importantly, this challenge extends to ARM when applied to probabilistic models operating in continuous domains [1]. These limitations can have negative impacts on both the effectiveness and efficiency of the robust MLE framework.

In this work, we leverage the Distributionally Robust Supervised Learning (DRSL) framework (Namkoong and Duchi, 2016; Hu et al., 2018) for learning robust TPMs in continuous domains. Specifically, we first show that the DRSL framework can be utilized to learn distributionally robust probabilistic models by designing a special loss function based on the negative log density of data points. We further demonstrate the efficiency of our approach by showing that the inner optimization problem can be solved exactly in

---

[1]Certain tractable models, with a static ordering of variables, that operate in discrete domains such as arithmetic circuits, PSDDs, etc. admit exact solutions in polynomial time (Peddi et al., 2022).

linearithmic time, while the outer optimization problem is equivalent to a standard MLE problem on weighted data. In essence, our approach offers an efficient way to equip probabilistic models with distributional robustness, while only requiring that the underlying probabilistic models admit tractable loglikelihood computation and efficient learning on weighted data.

This paper makes the following contributions:

1. We introduce a novel application of the DRSL framework for learning distributionally robust probabilistic models. This presents an important alternative for the development of robust probabilistic models.

2. We develop an efficient algorithm capable of finding the exact solution to the inner optimization objective of the adversarial minimization problem within the DRSL framework, when the KL-divergence is employed as the metric for measuring distributional distances. Moreover, we demonstrate that the outer optimization problem aligns with the standard MLE learning process on weighted data.

3. We conduct empirical evaluations on the proposed algorithm and methods by learning robust continuous TPMs and evaluating the loglikelihoods on both initial uncorrupted and adversarial test sets against their counterparts learned through the standard MLE framework across nine real-world datasets.

## 2 BACKGROUND

We use bold uppercase letters for a set of random variables, e.g., $\boldsymbol{X}$, while a single random variable is denoted using normal uppercase letters, e.g., $A$. In addition, members of a set of random variables will be indexed by subscripts, e.g., $X_i$ denotes the $i^{th}$ random variable in set $\boldsymbol{X}$. The size of set of random variables $\boldsymbol{X}$ is denoted as $|\boldsymbol{X}|$. The instantiations (configurations) of random variables are denoted as lowercase letters. For example, $\boldsymbol{x}$ is one possible configuration for all variables in $\boldsymbol{X}$ and $a$ is a possible value that the random variable $A$ can take. If $\boldsymbol{Y}$ is a subset of $\boldsymbol{X}$, then the projection of assignment $\boldsymbol{x}$ onto set $\boldsymbol{Y}$ is denoted as $\boldsymbol{x_Y}$. All random variables considered in this paper are assumed to be real-valued unless otherwise noted.

### 2.1 DISTRIBUTIONALLY ROBUST SUPERVISED LEARNING

In this section, we first introduce the Empirical Risk Minimization (ERM) framework for supervised learning along with its connection to Maximum Likelihood Estimation (MLE). We then describe the Distributionally Robust Supervised Learning (DRSL) framework, which incorporates distributional robustness into the ERM framework and show

that the DRSL framework can be employed as a surrogate for MLE to learn robust probabilistic models.

#### 2.1.1 Empirical Risk Minimization (ERM)

In a typical supervised learning setting, the input training data is assumed to be free of corruption and noise, and our objective is to find the model parameters, $\theta$, that minimize the expected loss with respect to the unknown data distribution, i.e.,

$$\underset{\theta}{\operatorname{argmin}} \, \mathbb{E}_{(\boldsymbol{x},y) \sim P(\boldsymbol{X},Y)}[\mathcal{L}(\boldsymbol{x}, y, \theta)].$$

Here, $P(\boldsymbol{X}, Y)$ is the unknown data distribution over the input feature variables $\boldsymbol{X}$ and label variable $Y$; and $\mathcal{L}(\boldsymbol{x}, y, \theta)$ is the loss function. Given a dataset $\mathcal{D} = \{(x_i, y_i) | i = 1, ..., n\}$ that is assumed to consist of i.i.d samples drawn from the distribution $P(\boldsymbol{X}, Y)$, the above expectation optimization problem can be approximated as

$$\theta^* = \underset{\theta}{\operatorname{argmin}} \frac{1}{n} \sum_{i=1}^{n} \mathcal{L}_i(\theta),$$

where $\mathcal{L}_i(\theta) \equiv \mathcal{L}(\boldsymbol{x_i}, y_i, \theta)$ is the loss respect to the $i^{th}$ input data instance.

#### 2.1.2 Maximum Likelihood Estimation (MLE) as ERM

Given data observations $\mathcal{D} = \{\boldsymbol{z}_1, ..., \boldsymbol{z}_n\}$ that are drawn independently from distribution $P_\theta(Z)$ with unknown parameter $\theta$, the goal of MLE is to find the parameters that maximize the log-likelihood, i.e.,

$$\underset{\theta}{\operatorname{argmax}} \log \mathcal{L}_\theta(\mathcal{D}) = \underset{\theta}{\operatorname{argmax}} \sum_{i=1}^{n} \log P_\theta(\boldsymbol{z_i}).$$

This can be put into the ERM framework by setting the loss function as the negative log-likelihood, $\mathcal{L}_i(\theta) = -\log P_\theta(\boldsymbol{z_i})$, and observing that

$$\theta^* = \underset{\theta}{\operatorname{argmax}} \sum_{i=1}^{n} \log P_\theta(\boldsymbol{z_i}) = \underset{\theta}{\operatorname{argmin}} \frac{1}{n} \sum_{i=1}^{n} \mathcal{L}_i(\theta),$$

which means we can use ERM framework to solve the MLE task. Note that scaling the objective function by $1/n$ doesn't change the optimal parameters, $\theta^*$.

#### 2.1.3 DRSL formulation

Unlike ERM, DRSL (Bauso et al., 2017; Namkoong and Duchi, 2016) is explicitly formulated for the cases where the test distribution $Q$ is shifted from the training distribution $P$. Specifically, DRSL considers the test distribution $Q$ from

an uncertainty set $U_{P,\delta}$ that contains all distributions within a $\delta$ f-divergence from the distribution $P$, i.e.,

$$U_{P,\delta} = \{Q|D_f(Q,P) \le \delta\},$$

where $D_f(Q,P) = \mathbb{E}_P\left[f\left(Q/P\right)\right]$ is the f-divergence between distribution $Q$ and $P$, with a convex function $f(\cdot)$ that satisfies $f(1) = 0$ ($P = Q$ implies zero distance). Note that the support of distribution $Q$ is assumed to be a subset of the support of distribution $P$. In other words, $P(x) = 0$ implies $Q(x) = 0$. When $f(x) = x \log x$, the f-divergence reduces to the well-known Kullback-Leibler divergence (Kullback and Leibler, 1951). The $\delta$ in the above equation is a hyper-parameter that controls the amount of distributional shift. When $\delta = 0$, DRSL reverts to standard ERM learning.

The objective of DRSL is to find the best parameter $\theta$ that minimizes the expected loss with respect to the worst test distribution $Q \in U_{P,\delta}$, and it can be formulated as a minimax problem as follows.

$$\underset{\theta}{\arg\min} \sup_{Q \in U_{P,\delta}} \mathbb{E}_{(\boldsymbol{x},y)\sim Q(\boldsymbol{X},Y)}[\mathcal{L}(\boldsymbol{x},y,\theta)]$$

Setting $r(\boldsymbol{x},y) = \frac{Q(\boldsymbol{x},y)}{P(\boldsymbol{x},y)}$, we can reformulate the objective as

$$\underset{\theta}{\arg\min} \sup_{r \in \mathcal{U}_{P,\delta}} \mathbb{E}_{(\boldsymbol{x},y)\sim P(\boldsymbol{X},Y)}[r(\boldsymbol{x},y)\mathcal{L}(\boldsymbol{x},y,\theta)],$$

where

$$\mathcal{U}_{P,\delta} = \{r(\boldsymbol{x},y)|\mathbb{E}_{P(\boldsymbol{X},Y)}[f(r(\boldsymbol{x},y))] \le \delta,$$
$$\mathbb{E}_{P(\boldsymbol{X},Y)}[r(\boldsymbol{x},y)] = 1,$$
$$r(\boldsymbol{x},y) \ge 0\}.$$

The first constraint in the set $\mathcal{U}_{P,\delta}$ guarantees that the f-divergence between $Q$ and $P$ is less or equal to $\delta$ while the second and third constraints guarantee $Q$ is a valid distribution. Similar to the ERM case, the expectations in the above formulation can be approximated using samples as

$$\underset{\theta}{\arg\min} \sup_{r \in \hat{\mathcal{U}}_\delta} \frac{1}{n}\sum_{i=1}^{n} r_i \cdot \mathcal{L}_i(\theta), \tag{1}$$

where

$$\hat{\mathcal{U}}_\delta = \left\{ r \mid \frac{1}{n}\sum_i f(r_i) \le \delta, \frac{1}{n}\sum_i r_i = 1, r_i \ge 0 \right\},$$

$r_i = r(\boldsymbol{x_i}, y_i)$, and $r = (r_1, r_2, ..., r_n)$ is the vector of density ratios. This problem can be treated as a minimax game between an adversary and a learner in which the adversary reweights the losses of all data instances using $r$, and the learner then tries to minimize the weighted loss (Hu et al., 2018; Bauso et al., 2017).

As discussed in Section 2.1.2, by choosing the loss function as the negative log-likelihood, $\mathcal{L}_i(\theta) = -\log P_\theta(\boldsymbol{z_i})$, and

plugging it into (1), we can use the DRSL framework to learn a robust probabilistic model $P_\theta(\boldsymbol{Z})$. In fact, as we will shown later in Section 3, the inner maximization problem can be solved exactly in linearithmic time when KL-divergence is employed – assuming that the loglikelihoods $\log P_\theta(\boldsymbol{z})$ can be computed efficiently and the probabilistic model admits efficient learning on weighted data.

## 3 METHODOLOGY

In this section, we present our approach to learn distributionally robust probabilistic models, leveraging the DRSL framework with the KL-divergence as the distance metric between distributions. Specifically, given a set of data observations $\mathcal{D} = \{\boldsymbol{z}_1, ..., \boldsymbol{z}_n\}$ that are sampled independently from a distribution $P_\theta(\boldsymbol{Z})$ with unknown parameter $\theta$, our goal is to find the best parameters such that the adversarial "risk" in (1) is minimized. Formally, the problem is expressed as follows.

$$\underset{\theta}{\arg\max} \inf_{r} \frac{1}{n}\sum_{i=1}^{n} r_i \cdot \log P_\theta(\boldsymbol{z_i})$$
$$s.t. \tag{2}$$
$$\frac{1}{n}\sum_{i=1}^{n} r_i \log r_i \le \delta, \quad \frac{1}{n}\sum_{i=1}^{n} r_i = 1, \quad r_i \ge 0$$

The inner optimization problem ($\inf_r$) corresponds to an *adversarial* step where we find the worst weight $r$ such that the reweighted loglikelihood is minimized; while the outer optimization problem ($\arg\max_\theta$) corresponds to a *learning* step where we find the best $\theta$ to maximize the weighted loglikelihood. Taking the same approach as Generative Adversarial Networks (GAN) (Goodfellow et al., 2020), we tackle this optimization problem by alternating between the learning and adversarial steps as follows.

1. **Init**: Initialize the parameters $\theta$ of model $P_\theta(\boldsymbol{Z})$.

2. **Adversarial Step**: Fix $\theta$, update the weight vector $r$ by solving the inner minimization problem.

3. **Learning Step**: Fix $r$, update the parameter $\theta$ by solving the outer maximization problem.

4. **Repeat**: Repeat step 2-3 until a suitable stopping condition is met, such as reaching a stationary point or hitting the maximum number of iterations.

The following sections provide a detailed prescription for solving the optimization problems in both the adversarial and learning steps. Subsequently, we will demonstrate how it can be employed to train robust probabilistic models.

## 3.1 ADVERSARIAL STEP

The optimization problem we need to solve in the adversarial step is

$$\inf_{\boldsymbol{r}} \sum_{i=1}^{n} r_i \cdot l_i$$

$$s.t. \tag{3}$$

$$\frac{1}{n}\sum_{i=1}^{n} r_i \log r_i \leq \delta, \quad \frac{1}{n}\sum_{i=1}^{n} r_i = 1, \quad r \geq 0,$$

where $l_i \equiv \log P_\theta(\boldsymbol{z_i})$ can be treated as a constant because the parameter $\theta$ is fixed. In addition, we also ignore the constant $1/n$ in the objective function because it doesn't change the optimal weight values, $\boldsymbol{r}$.

We employ the method of Lagrange multipliers and derive the dual problem of the optimization problem in (3) as

$$\sup_{\alpha \geq 0, \beta} \quad -\sum_i \alpha \cdot \exp\left(\frac{-\beta - l_i}{\alpha} - 1\right) - \alpha n\delta - \beta n, \tag{4}$$

where $\alpha, \beta$ are Lagrange multipliers (detailed derivation is shown in Appendix Section A) and the value of primal variable $r_i$ can be calculated as

$$r_i = \exp\left(\frac{-\beta - l_i}{\alpha} - 1\right). \tag{5}$$

Note that the primal problem satisfies the Slater's condition (Slater, 2013), therefore ensuring strong duality.

We can attempt to solve (4) by taking the derivatives respective to $\alpha$ and $\beta$ and setting them equal to zero (assuming there is a non-negative solution for $\alpha$). Denoting the objective in (4) as $L'(\alpha, \beta)$, we have

$$\frac{\partial L'}{\partial \alpha} = -n\delta + \sum_i \exp\left(\frac{-\beta - l_i}{\alpha} - 1\right)\left(\frac{-\beta - l_i}{\alpha} - 1\right) = 0 \tag{6}$$

and

$$\frac{\partial L'}{\partial \beta} = -n + \sum_i \exp\left(\frac{-\beta - l_i}{\alpha} - 1\right) = 0. \tag{7}$$

Because $r_i = \exp\left(\frac{-\beta - l_i}{\alpha} - 1\right)$, we can see that (6) is equivalent to $\sum_i r_i \log r_i = n\delta$; and (7) is equivalent to $\sum_i r_i = n$. These two equations correspond to our original constraints, and no closed form solution for $\alpha$ and $\beta$ is available.

Nevertheless, we note that the dual problem involves only two variables, $\alpha \geq 0$ and $\beta$, and the dual objective function in (4) derived from the method of Lagrange multipliers is always concave. We further prove that the dual objective is twice differentiable and *strictly* concave unless all loglikelihoods, $l_i$, are equal, which is unlikely given that $l_i$ is real-valued and there are usually many training instances

---

**Algorithm 1:** Efficient Linearithmic Search Algorithm for the Adversarial Step

**Input**: (1) $\boldsymbol{l} = \{l_1, ..., l_n\}$, the loglikelihoods with respect to all training data points (under the current model parameter $\theta$); (2) U, upper searching bound of the dual variable $\alpha$; (3) $\delta$, the hyper-parameter controlling the amount of distribution shifts; and (4) $\epsilon$, the maximum allowed error for the variable $\alpha$.

**Output**: the weight vector $\boldsymbol{r}$ that solves the optimization problem (3).

$\alpha_l \leftarrow 0$ ;
$\alpha_u \leftarrow U$ ;
**while** $\alpha_u - \alpha_l \geq \epsilon$ **do**
  $\quad \alpha \leftarrow (\alpha_l + \alpha_u)/2$ ;
  $\quad \beta \leftarrow -\alpha \log\left(\frac{n}{\sum_i \exp(-l_i/\alpha - 1)}\right)$ ;
  $\quad \boldsymbol{r} \leftarrow \exp\left(\frac{-\beta - \boldsymbol{l}}{\alpha} - 1\right)$ ;
  $\quad g \leftarrow -n\delta + \sum_i \log r_i^{r_i}$ //numerical stability ;
  $\quad$ **if** $g \leq 0$ **then**
    $\quad\quad$ // negative gradient
    $\quad\quad \alpha_u \leftarrow \alpha$ ;
  $\quad$ **else**
    $\quad\quad \alpha_l \leftarrow \alpha$ ;
  $\quad$ **end**
**end**
Return the weight vector $\boldsymbol{r}$ ;

---

(see Appendix Section B for a detailed proof). This implies that numerical optimization algorithms like gradient ascent or coordinate ascent can be employed to arbitrarily well approximate a *globally* optimal solution with enough iterations (Tseng, 2001).

We propose an efficient search-based algorithm capable of solving the problem and obtaining a solution that is arbitrarily close to the exact answer. Our algorithm is essentially a coordinate ascent algorithm with two key changes that guarantee linearithmic time complexity.

1. When conducting coordinate ascent along the $\beta$ direction ($\alpha$ is fixed), we can solve for $\beta$ in closed form using (7) as

$$\beta = -\alpha \log\left(\frac{n}{\sum_i \exp\left(\frac{-l_i}{\alpha} - 1\right)}\right) \tag{8}$$

2. When optimizing along the $\alpha$ direction with $\beta$ fixed, we use binary search (highly efficient with guaranteed logarithmic time complexity) instead of gradient ascent because it is a *one-dimensional strictly concave* maximization problem.

With the above key observations, we can effectively binary search for the optimal $\alpha$ using the method shown in Algorithm 1. Specifically, we first fix the $\alpha$ as the middle point

of the interval $[\alpha_l, \alpha_u]$, then compute the corresponding $\beta$ using the equation (8), and use the derivative evaluated at this point to constrain the location of the optimal $\alpha$. Note that the initial upper bound $\alpha = U$ must have a negative derivative, i.e., (6), when evaluated at the corresponding optimal $\beta$. One of the approaches to obtain such a upper bound is to keep multiplying $U$ by two until the gradient becomes negative. In addition, we calculate $\log r_i^{T_i}$ as the surrogate for $r_i \log r_i$ for numerical stability.

The time complexity for Algorithm 1 is $O(n \log(U/\epsilon))$ where $n$ is the number of training instances, $U$ is the search upper bound and $\epsilon$ is the error tolerance. The algorithm is very efficient even if $U$ is very large and we have high accuracy requirement. For example, when $U = 10^{20}$ (larger than the biggest 64-bit integer) and $\epsilon = 10^{-5}$, $\log(U/\epsilon)$ is less than 84. In practice, because the absolute loglikelihoods $|l_i|$ are usually small, the value of $U$ is small as well. Therefore, $\log(U/\epsilon)$ is typically less than 30.

### 3.2 LEARNING STEP

The optimization problem in the learning step is

$$\operatorname*{argmax}_{\theta} \sum_{i=1}^{n} r_i \cdot \log P_\theta(\boldsymbol{z_i}), \qquad (9)$$

where $r_i$ is a fixed constant weight. In fact, the above problem is equivalent to learning a model via standard MLE in which each data instance $\boldsymbol{z_i}$ is associated with a weight $r_i$ (Legeleux et al., 2022). Here, the weight can be interpreted as "how many times we see the data instance" [2]. Therefore, given a data instance $\boldsymbol{z_i}$ that is observed $r_i$ times, the loglikelihood with respect to this instance can be formulated as

$$\log P_\theta(\boldsymbol{z_i})^{r_i} = r_i \log P_\theta(\boldsymbol{z_i}),$$

which corresponds to the weighted loglikelihood in (9).

In general, the weighted MLE problem can be solved using the gradient ascent method [3] similar to the standard MLE case. In addition, for certain probabilistic models such as Multivariate Gaussians, this problem admits a closed form solution.

### 3.3 PRACTICAL CONCERNS

The adversarial step is usually significantly faster (typically around 1-5 seconds), compared to the learning step that often takes minutes or even hours due to the iterative EM or gradient optimization processes over neural networks.

---

[2]Float values are allowed here and it won't break the evaluation of joint likelihood.

[3]We limit our focus to model parameters during the learning step, assuming the model's structure is fixed (if the model involves a structural learning component).

Table 1: Number of instances and features of nine datasets (after preprocessing).

| Name | #instance | #feature |
|------|-----------|----------|
| airquality | 9357 | 12 |
| energy | 19735 | 24 |
| hepmass | 150000 | 21 |
| miniboone | 36488 | 43 |
| onlinenews | 39644 | 32 |
| parkinson | 5875 | 15 |
| sdd | 58509 | 29 |
| superconduct | 21263 | 68 |
| mnist (d20) | 70000 | 20 |

Therefore, executing EM or NN gradient updates until convergence in the learning step to achieve an accurate estimation of equation (9) would be highly inefficient. It is also essential to recognize that running EM to convergence is likely unnecessary, given that, in early rounds, the current weight vector $\boldsymbol{r}$ is sub-optimal and will be changed in subsequent iterations.

Therefore, a practical strategy is to execute the learning step for only a few iterations, aiming to identify a good or moderate parameter configuration under the current weight settings. After that, we promptly transition to the adversarial step to update the weight vector. This approach enhances the efficiency of the entire adversarial learning process and enables multiple iterations between the adversarial and learning steps.

## 4 EXPERIMENTS

In this section, we present empirical evaluations of the proposed method in Section 3 for learning distributionally robust probabilistic models in continuous domains.

### 4.1 EXPERIMENT SETUP

We consider nine real-world datasets in our experiments. One of them is the MNIST image dataset (LeCun et al., 1998), while the other eight datasets are selected from the UCI machine learning repository (Dua and Graff, 2017). Following Uria et al. (2016), we preprocess all UCI datasets by eliminating discrete valued features and one of the attributes from every pair of attributes whose Pearson correlation coefficient is greater than $0.98$. For the MNIST dataset, we train a variational auto encoder (Kingma and Welling, 2013) and embed each input image as a 20 dimensional feature vector in a structured hidden Gaussian space [4]. All datasets were normalized by subtracting the mean and then dividing by the

---

[4]The encoder and decoder architecture are based on convolutional neural networks (CNNs).

Table 2: Loglikelihood scores, average test LL scores and number of wins (ties are ignored) of robust and standard MixMG models on various test sets and neighboring regions. The robust model learned through our algorithm achieved higher or similar average LL scores on most of the adversarial cases.

| Dataset | Method | Original Test | Gaussian Test | Jittter Test | Worst NB | Average NB |
|---------|--------|---------------|---------------|--------------|----------|------------|
| airquality | DRSL | 9.29 ± 4.6 | -274.58 ± 201.1 | -322.60 ± 1378.4 | -290.95 ± 133.7 | -104.09 ± 74.6 |
| | MLE | 9.53 ± 6.3 | -347.31 ± 205.3 | -879.70 ± 4533.5 | -380.05 ± 130.6 | -137.12 ± 74.4 |
| energy | DRSL | -7.55 ± 5.3 | -32.03 ± 17.0 | -63.25 ± 64.3 | -50.29 ± 20.2 | -21.92 ± 7.7 |
| | MLE | -6.58 ± 6.6 | -39.75 ± 25.6 | -91.09 ± 98.3 | -67.26 ± 29.7 | -23.71 ± 9.4 |
| hepmass | DRSL | -25.15 ± 4.9 | -29.24 ± 3.9 | -26.08 ± 4.6 | -31.31 ± 2.9 | -27.06 ± 4.1 |
| | MLE | -24.36 ± 4.7 | -28.52 ± 4.6 | -25.46 ± 4.9 | -30.89 ± 4.2 | -26.31 ± 4.3 |
| miniboone | DRSL | -24.28 ± 15.6 | -43.80 ± 14.0 | -53.85 ± 18.5 | -51.96 ± 12.3 | -32.67 ± 13.1 |
| | MLE | -21.41 ± 14.8 | -44.58 ± 13.9 | -56.18 ± 20.9 | -53.59 ± 11.6 | -31.38 ± 12.8 |
| mnist | DRSL | -3.57 ± 6.0 | -6.01 ± 2.3 | -6.56 ± 2.3 | *N/A* | *N/A* |
| | MLE | -0.59 ± 6.4 | -4.41 ± 4.2 | -5.87 ± 4.0 | *N/A* | *N/A* |
| onlinenews | DRSL | -1.61 ± 19.2 | -240.22 ± 229.9 | -1009.90 ± 1403.0 | -627.79 ± 84.8 | -115.92 ± 17.2 |
| | MLE | -1.22 ± 27.5 | -257.00 ± 237.3 | -1013.92 ± 1404.9 | -632.71 ± 99.9 | -120.18 ± 53.1 |
| parkinson | DRSL | -5.47 ± 7.1 | -14.30 ± 5.9 | -15.88 ± 10.8 | -18.83 ± 4.7 | -10.07 ± 5.1 |
| | MLE | -3.81 ± 11.1 | -16.13 ± 9.9 | -20.47 ± 33.9 | -21.20 ± 13.3 | -10.57 ± 9.9 |
| sdd | DRSL | 0.62 ± 42.7 | -95.17 ± 39.1 | -65.48 ± 37.9 | -95.49 ± 39.4 | -86.71 ± 39.5 |
| | MLE | -3.83 ± 118.9 | -55.58 ± 116.6 | -50.20 ± 797.6 | -57.80 ± 120.0 | -53.03 ± 117.9 |
| superconduct | DRSL | 59.43 ± 49.6 | -235.21 ± 88.3 | -880.96 ± 968.7 | -249.90 ± 59.7 | -101.63 ± 34.3 |
| | MLE | 62.82 ± 52.2 | -384.21 ± 140.7 | -1480.38 ± 1632.5 | -394.99 ± 94.1 | -164.14 ± 54.3 |
| Average | DRSL | 0.19 | **-107.84** | **-271.62** | **-177.06** | **-62.51** |
| | MLE | **1.17** | -130.83 | -402.59 | -204.81 | -70.81 |
| #Wins | DRSL | 1 | **6** | **6** | **5** | **4** |
| | MLE | **8** | 3 | 3 | 1 | 2 |

standard deviation. The number of instances and features for each dataset after preprocessing is shown in Table 1. Note that for the eight UCI datasets, the train/test split is not defined from the data source, and we randomly chose 85% of the data instances for the training split and the remaining were used to form the test split. We further set aside 20% of the training instances for validation purposes.

We consider the following two types of probabilistic models in our experiment.

1. Mixture of Multivariate Gaussian (MixMG), which serves as a standard benchmark model. The number of mixture components is treated as a hyper-parameter and is automatically tuned from the range of three to nine. Note that fitting a MixMG model on weighted data can be conducted efficiently using the EM algorithm where the solution for the E-step and M-step are still in closed form (Legeleux et al., 2022).

2. NN-GBN model proposed by Dong et al. (2022) [5], which models the full joint distribution as the product of a local, complex distribution over a small subset of variables and a fully tractable conditional distribution whose parameters are controlled using a neural

network. We choose this model for case study simply because (1) we are interested in continuous domains; and (2) NN-GBN can be easily adapted for parameter learning on weighted data by adding an extra weight term into the original loss function and return the weighted negative loglikelihood as the loss. We tune the following two hyper-parameters for NN-GBN: (1) the maximum learning rate from the set $\{10^{-2}, 3.3 \times 10^{-3}, 10^{-3}\}$; and (2) the weight decay from the set $\{10^{-3}, 10^{-4}\}$. Additionally, we employed the OneCycleLR scheduler with cosine decay in PyTorch (Paszke et al., 2019) to manage the learning rate. All other training configurations remain unchanged and align with what used in the original code and paper.

For each type of the probabilistic model, we first learn a robust model through the algorithmic methodology described in Section 3, and then compare its performance against the standard model learned using the MLE framework. We conduct 150 iterations of learning and adversarial steps, and the optimal hyperparameter is selected based on the model that has highest log-likelihood achieved on the validation set. In addition, following Hu et al. (2018), we set the maximum amount of distribution shift $\delta = 0.5$.

As discussed in Section 3.3, conducting the learning step with full iterations is inefficient and unnecessary. Therefore, we only perform one training epoch of the neural network

---

[5]The model introduced by the authors remains unnamed in their publication, and for the purposes of this study, we will refer to it as NN-GBN.

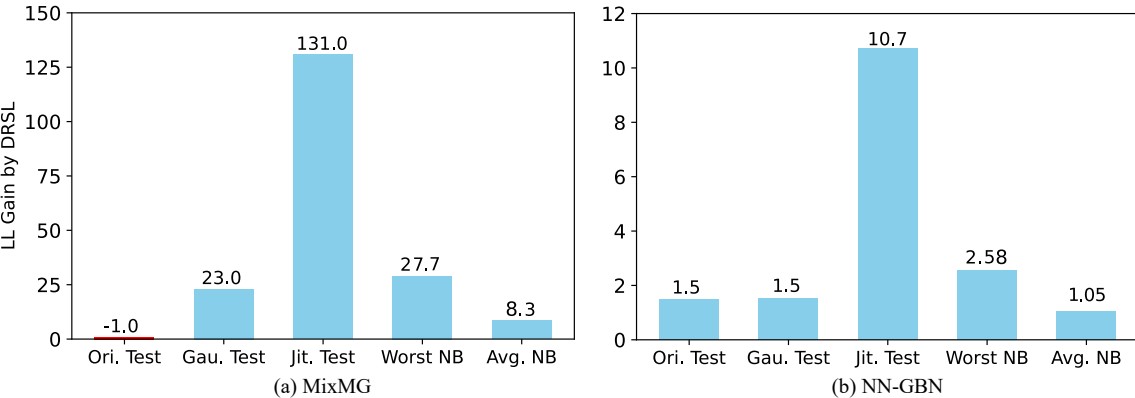

Figure 1: The average amount of loglikelihood improved through DRSL robust learning for both (a) MixMG and (b) NN-GBN models in all five test scenarios.

per learning step for the NN-GBN model. To encourage convergence, we initiate the neural network with 50 training epochs on unweighted training data to prevent situations in which inaccurately estimated log-likelihood negatively affects the early adversarial iterations, potentially resulting in poor weight assignments for our data. Such poor weights could adversely impact the learning process and ultimately lead to divergence.

All experiments were conducted on a workstation equipped with a 16-core Intel Xeon Gold 6130 CPU and two Quadro P5000 GPUs. The datasets and codes used in the experiment are publicly available on Github [6].

## 4.2 ADVERSARIAL GENERATIVE PERFORMANCE

We evaluated the generative performance of the robust model against its standard counterparts for both MixMG and NN-GBN models, by comparing their average loglikelihood [7] achieved on the original uncorrupted test set as well as two additional adversarial test sets for each dataset. To be more specific, we create two types of adversarial test sets for each dataset as follows.

1. Adding Gaussian noise to the original test set, the Gaussian distribution used in our experiment is with zero mean and standard deviation one. In addition, we clip all the noise values to the interval $[-0.2, 0.2]$ to simulate the scenario where we have minor or medium level perturbation of the data.

2. Jitter the input by setting some entries to a random

number (Su et al., 2019). In our experiment, we randomly pick 20% of the input entries and assign a value uniformly sampled from the interval $[-0.2, 0.2]$. This is considered to be a harder task because the amount of value change can be far greater than the previous case.

We also investigated how the learned distribution behaves around the test points: we first randomly sampled 500 instances around each test point and assessed the loglikelihoods of these neighboring data points. Subsequently, we identified the instance with the lowest loglikelihood among these neighbors (denoted as the 'Worst NB') and computed the average loglikelihood for these 500 points (denoted as 'Average NB'). These two resulting values constitute the neighbor metrics for the test point. We repeated this process and calculated the average of these two metrics over all test points within each dataset. Note that for the MNIST dataset, because the inputs are embedding vectors from the trained variational autoencoder, the neighboring vectors may not correspond to any real image inputs. For this reason, we exclude the results of MNIST neighboring data from the loglikelihood calculation and the summarization process.

We report the test set loglikelihoods and corresponding standard deviations achieved for MixMG and NN-GBN in Tables 2 and 3, respectively. Furthermore, we have summarized the performance improvements regarding the average loglikelihoods of the robust model compared to its standard counterparts in Figure 1 for all five types of testing scenarios. Note that for the NN-GBN model, both robust and standard models achieved identical results for the miniboone dataset. This consistency arises from our approach of continuously monitoring the models' performance on the validation set at each iteration and retaining the model in its optimal state. For the miniboone dataset, which is particularly susceptible to overfitting, both training methods delivered their best performance during the early pre-training iterations.

From these results, we have the following observations. Firstly, the model trained with the DRSL framework consis-

---

[6]UAI2024-RobustLearning

[7]Generally speaking, loglikelihood is not a good metric for evaluating a model's generative performance in continuous domains because it can be unbounded (Dong et al., 2022), unless the models being compared are in the same parametric family (which is the case for us).

Table 3: Loglikelihood scores, average test LL scores and number of wins (ties are ignored) of robust and standard NN-GBN models on various test sets and neighboring regions. The robust model achieved higher average LL scores on all test cases including the original uncorrupted test set.

| Dataset | Method | Original Test | Gaussian Test | Jittter Test | Worst NB | Average NB |
|---|---|---|---|---|---|---|
| airquality | DRSL | -1.16 ± 9.3 | -26.42 ± 68.9 | -218.76 ± 3270.2 | -14.17 ± 38.1 | -3.75 ± 11.2 |
| | MLE | -1.23 ± 8.3 | -25.50 ± 64.1 | -311.49 ± 5217.5 | -13.42 ± 28.2 | -3.65 ± 9.1 |
| energy | DRSL | 0.99 ± 10.4 | -66.97 ± 63.1 | -142.42 ± 179.0 | -41.02 ± 40.0 | -5.94 ± 10.6 |
| | MLE | 0.99 ± 10.8 | -67.96 ± 63.7 | -142.68 ± 175.2 | -41.96 ± 41.7 | -6.02 ± 10.9 |
| hepmass | DRSL | -26.81 ± 5.4 | -27.92 ± 6.3 | -26.52 ± 4.6 | -28.44 ± 6.2 | -26.91 ± 5.4 |
| | MLE | -26.81 ± 5.4 | -27.92 ± 6.3 | -26.50 ± 4.6 | -28.44 ± 6.2 | -26.91 ± 5.4 |
| miniboone | DRSL | -26.30 ± 17.9 | -48.40 ± 20.0 | -78.54 ± 101.3 | -38.91 ± 21.2 | -28.43 ± 17.2 |
| | MLE | -26.30 ± 17.9 | -48.40 ± 20.0 | -78.54 ± 101.3 | -38.91 ± 21.2 | -28.43 ± 17.2 |
| mnist | DRSL | -10.33 ± 6.2 | -13.22 ± 6.0 | -13.01 ± 5.5 | *N/A* | *N/A* |
| | MLE | -10.31 ± 6.2 | -13.20 ± 5.8 | -13.02 ± 5.5 | *N/A* | *N/A* |
| onlinenews | DRSL | -19.43 ± 63.6 | -44.63 ± 129.2 | -39.19 ± 88.0 | -35.45 ± 114.5 | -23.11 ± 66.2 |
| | MLE | -20.40 ± 107.3 | -59.34 ± 338.5 | -44.96 ± 161.9 | -50.66 ± 283.8 | -25.81 ± 115.1 |
| parkinson | DRSL | -5.05 ± 6.9 | -16.89 ± 9.7 | -25.87 ± 50.2 | -12.21 ± 8.3 | -6.20 ± 6.7 |
| | MLE | -5.07 ± 7.4 | -17.69 ± 10.6 | -28.35 ± 53.0 | -13.02 ± 9.5 | -6.32 ± 7.2 |
| sdd | DRSL | -20.40 ± 490.1 | -56.76 ± 491.1 | -39.50 ± 514.8 | -52.93 ± 493.2 | -34.04 ± 489.9 |
| | MLE | -36.21 ± 1922.4 | -66.12 ± 1907.2 | -46.96 ± 1381.2 | -64.81 ± 1934.7 | -48.56 ± 1921.7 |
| superconduct | DRSL | 40.44 ± 45.4 | -219.83 ± 105.0 | -756.24 ± 702.0 | -56.08 ± 59.6 | 7.11 ± 37.2 |
| | MLE | 43.79 ± 45.6 | -208.76 ± 102.9 | -744.03 ± 702.2 | -48.62 ± 62.2 | 16.01 ± 41.4 |
| Average | DRSL | **-7.56** | **-57.89** | **-148.90** | **-34.90** | **-15.16** |
| | MLE | -9.06 | -59.43 | -159.61 | -37.48 | -16.21 |
| #Wins | DRSL | **4** | **4** | **5** | **3** | **3** |
| | MLE | 2 | 3 | 2 | 1 | 1 |

tently outperformed its counterpart in terms of the average loglikelihood score for both adversarial test sets, as shown in Figure 1. We noted a particularly intriguing result: models trained using the DRSL framework also achieved higher or similar average loglikelihood scores on the original, uncorrupted test set at times. Several factors may contribute to this phenomenon: (1) the model learned through DRSL framework shapes the distribution more effectively, rather than spreading densities over neighbors, which tends to lead to lower loglikelihood on the original test set in practice; (2) the model tuning process prioritizes performance on the original validation set, emphasizing the importance of focusing on both adversarial examples and the original data.

Secondly, we observed that DRSL exhibited more substantial improvements on the jittered adversarial data compared to the Gaussian adversarial test set. This suggests that distributionally robust learning offers higher tolerance for challenging corruptions such as measurement errors.

Lastly, from the results on the two neighboring metrics, it becomes evident that the robust model excels in shaping a distribution that exhibits a high degree of smoothness around real data points. In practical terms, this suggests that DRSL effectively captures the underlying data distribution, ensuring that it doesn't just account for isolated, exceptional cases, but rather models the broader data context more comprehensively and robustly. This characteristic contributes to

its superior performance across the different testing scenarios considered in our experiments. We additionally note that our robust model has shown more significant improvements on the worst neighbor metric compared to the average neighbor metric. This observation closely aligns with our earlier findings, where the performance enhancement in the jitter test set is higher when compared to the Gaussian test set.

## 5 CONCLUSION

In this work, we presented a novel approach for learning distributionally robust tractable probabilistic models (TPMs) through the DRSL framework. We proposed valuable insights and efficient algorithms for addressing the adversarial optimization problems within the DRSL framework. Our empirical evaluations revealed an intriguing result: the models trained using the DRSL framework exhibited comparable or even superior performance on both adversarial and original uncorrupted test data.

We focused solely on parameter learning while assuming the underlying model has no structural learning requirement or the structure remains fixed during the robust learning steps. In future research, we aim to develop efficient structure learning algorithms on weighted data and extend this framework to other continuous TPMs such as Sum Product Networks (SPNs).

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

# Learning Distributionally Robust Tractable Probabilistic Models in Continuous Domains
# (Supplementary Material)

**Hailiang Dong**[1]   **James Amato**[1]   **Vibhav Gogate**[1]   **Nicholas Ruozzi**[1]

[1]Computer Science Dept., The University of Texas at Dallas, Texas, 75080, USA

## A   DERIVATION OF LAGRANGE MULTIPLIER METHOD FOR ADVERSARIAL STEP

The problem we have in the adversarial step is

$$\inf_{\boldsymbol{r}} \sum_{i=1}^{n} r_i \cdot l_i$$

$$s.t.$$

$$\frac{1}{n} \sum_{i=1}^{n} r_i \log r_i \leq \delta, \quad \frac{1}{n} \sum_{i=1}^{n} r_i = 1, \quad r_i \geq 0,$$

where $l_i \equiv \log P_\theta(\boldsymbol{z_i})$ can be treated as a constant because the parameter $\theta$ is fixed. In addition, we also ignore the constant $1/n$ in the objective function because it doesn't change the optimal weight values $\boldsymbol{r}$.

We begin by constructing the Lagrangian,

$$L(\boldsymbol{r}, \alpha, \beta) = \sum_i r_i l_i + \alpha \left( \sum_i r_i \log r_i - n\delta \right) + \beta \left( \sum_i r_i - n \right),$$

where $\alpha > 0$ and $\beta$ are Lagrange multipliers. Note that we omit the third constraint $r_i \geq 0$ in the above formulation because $r_i \log r_i$ already implies the constraint, and we will also show it is safe to drop it in the later part of the derivation. Take the derivative [1] of the Lagrange respect to $r_i$ and set the derivative to zero, we have

$$\frac{\partial L}{\partial r_i} = l_i + \alpha(\log r_i + 1) + \beta = 0,$$

and this gives us

$$r_i = \exp \left( \frac{-\beta - l_i}{\alpha} - 1 \right),$$

which is always greater than $0$ and this is also why we can safely drop the third constraint $r_i \geq 0$. Plugging the above equation back into the Lagrangian, we get our dual objective function as

$$L'(\alpha, \beta) = - \sum_i \alpha \cdot \exp \left( \frac{-\beta - l_i}{\alpha} - 1 \right) - \alpha n\delta - \beta n.$$

---

[1]The log function is of base $e$, i.e., the nature log.

Taking the derivative respect to $\alpha$ and $\beta$ and setting them to zero, we have

$$\frac{\partial L'}{\partial \alpha} = -n\delta - \sum_i \exp\left(\frac{-\beta - l_i}{\alpha} - 1\right) + \alpha \exp\left(\frac{-\beta - l_i}{\alpha} - 1\right)\frac{\beta + l_i}{\alpha^2}$$

$$= -n\delta - \sum_i \exp\left(\frac{-\beta - l_i}{\alpha} - 1\right)\left(\frac{\beta + l_i}{\alpha} + 1\right) = 0$$

(10)

and

$$\frac{\partial L'}{\partial \beta} = -n - \sum_i \alpha \cdot \exp\left(\frac{-\beta - l_i}{\alpha} - 1\right)\cdot -\frac{1}{\alpha}$$

$$= -n + \sum_i \exp\left(\frac{-\beta - l_i}{\alpha} - 1\right) = 0.$$

(11)

For detailed analysis and efficient algorithm, please refer to section 3.1.

## B  PROOF OF STRICT CONCAVENESS

In order to prove the dual objective function $L'$ is strictly concave, we need to show that the Hessian matrix (a $2 \times 2$ matrix in our case) is always negative-definite. To begin with, we first compute the Hessian matrix as follows.

$$A = \frac{\partial^2 L'}{\partial^2 \alpha} = \sum_i \exp\left(\frac{-\beta - l_i}{\alpha} - 1\right)\left(\frac{-\beta - l_i}{\alpha} - 1\right)\left(\frac{\beta + l_i}{\alpha^2}\right) + \exp\left(\frac{-\beta - l_i}{\alpha} - 1\right)\left(\frac{\beta + l_i}{\alpha^2}\right)$$

$$= \sum_i \exp\left(\frac{-\beta - l_i}{\alpha} - 1\right)\left(\frac{\beta + l_i}{\alpha^2}\right) * \left(\frac{-\beta - l_i}{\alpha} - 1 + 1\right)$$

$$= -\sum_i r_i \frac{(\beta + l_i)^2}{\alpha^3}$$

Here, we are using the fact that $r_i = \exp\left(\frac{-\beta - l_i}{\alpha} - 1\right)$. The other two derivatives are shown as follows.

$$C = \frac{\partial^2 L'}{\partial^2 \beta} = \sum_i \exp\left(\frac{-\beta - l_i}{\alpha} - 1\right) * -\frac{1}{\alpha}$$

$$= -\frac{1}{\alpha}\sum_i r_i$$

and

$$B = \frac{\partial^2 L'}{\partial \alpha \partial \beta} = \frac{\partial^2 L'}{\partial \beta \partial \alpha} = \sum_i \exp\left(\frac{-\beta - l_i}{\alpha} - 1\right) * \left(\frac{\beta + l_i}{\alpha^2}\right)$$

$$= \sum_i r_i \left(\frac{\beta + l_i}{\alpha^2}\right)$$

Therefore, the Hessian matrix is

$$M = \begin{bmatrix} A & B \\ B & C \end{bmatrix}.$$

To prove the above matrix $M$ is negative definite, we need to show the following two facts:

1. the trace $\text{trace}(M) = \lambda_1 + \lambda_2 < 0$, where $\lambda_1$ and $\lambda_2$ are the eigenvalues.
   **Proof:** the trace of matrix M is

$$A + C = -\sum_i r_i \frac{(\beta + l_i)^2}{\alpha^3} - \frac{1}{\alpha}\sum_i r_i.$$

Because $r_i = \exp\left(\frac{-\beta - l_i}{\alpha} - 1\right) > 0$ and $\alpha \geq 0$, the above formulate is always negative. Note that when $\alpha = 0$, the dual objective function is a constant and there is no need for optimization.

2. the determinant $\det(M) = \lambda_1 \cdot \lambda_2 > 0$.

   **Proof:** the determinant of matrix M is

   $$A \cdot C - B \cdot B = \left(\sum_i r_i \frac{(\beta + l_i)^2}{\alpha^3}\right)\left(\frac{1}{\alpha}\sum_i r_i\right) - \left(\sum_i r_i \left(\frac{\beta + l_i}{\alpha^2}\right)\right) \cdot \left(\sum_i r_i \left(\frac{\beta + l_i}{\alpha^2}\right)\right)$$

   $$= \frac{1}{\alpha^4}\left(\left(\sum_i r_i(\beta + l_i)^2\right) \cdot \left(\sum_i r_i\right) - \left(\sum_i r_i (\beta + l_i)\right) \cdot \left(\sum_i r_i (\beta + l_i)\right)\right).$$

   Denote

   $$X = \left(\sum_i r_i(\beta + l_i)^2\right) \cdot \left(\sum_i r_i\right)$$

   and

   $$Y = \left(\sum_i r_i (\beta + l_i)\right) \cdot \left(\sum_i r_i (\beta + l_i)\right),$$

   to further simplify the above equation, let's focus on the coefficients of item $r_i r_j, \forall i \leq j$ from $X$ and $Y$, respectively. Specifically, we have the coefficient of $r_i r_j$ in $X$ as

   $$(\beta + l_i)^2 + (\beta + l_j)^2,$$

   and the coefficient of $r_i r_j$ in $Y$ is

   $$2 \cdot (\beta + l_i) \cdot (\beta + l_j).$$

   And we have the difference between them as

   $$(\beta + l_i)^2 + (\beta + l_j)^2 - 2 \cdot (\beta + l_i) \cdot (\beta + l_j) = ((\beta + l_i) - (\beta + l_j))^2 = (l_i - l_j)^2.$$

   Therefore, we can simplify the determinant as

   $$A \cdot C - B \cdot B = \sum_i \sum_{j \geq i} \frac{(l_i - l_j)^2}{\alpha^4} r_i r_j \geq 0.$$

   It is easy to see that the determinant always greater than zero and is equal to zero only when the log-likelihoods of all training instances are equal, which is unlikely given that $l_i$ is real-valued and there are usually many training instances.

From above proof, we can conclude that, except the case that all log-likelihoods are equal (almost impossible in practice), both of the eigenvalue $\lambda_1$ and $\lambda_2$ are strictly negative, which means the matrix $M$ is negative definite and the objective function is strictly concave.