# OpenReview forum: "Learning Distributionally Robust Tractable Probabilistic Models in Continuous Domains"
_auai.org/UAI/2024/Conference — UAI 2024 poster_

### Official Review · Reviewer_cE6E · 2024-03-13

**Q2-1 Originality-Novelty:** 3
**Q2-2 Correctness-Technical Quality:** 3
**Q2-5 Clarity Of Writing:** 4

**Q1 Summary And Contributions:**

The paper presents a novel approach for acquiring a distributionally robust probabilistic model. This approach builds upon the DRSL framework, originally designed for supervised learning tasks, but here adapted for density estimation purposes. The core idea involves formulating the problem as a min-max optimization, wherein the outer optimization focuses on determining the parameters theta of a probability model, while the inner optimization seeks to identify a distribution shift within the training data distribution that maximally impacts the expected log likelihood loss. The author demonstrates that solving the inner optimization problem is achievable through solving a straightforward dual problem, while the outer optimization aligns with standard maximum likelihood estimation (MLE) learning but on weighted data. Furthermore, empirical evidence provided by the author showcases that models trained using this DRSL approach exhibit enhanced test likelihood compared to those obtained through MLE estimates. Additionally, when subjected to corruption in the test data, the parameters estimated by DRSL exhibit greater resilience on the corrupted test set.

**Q2-3 Extent To Which Claims Are Supported By Evidence:**

3: Good: the main claims are supported by convincing evidence (in the form of adequate experimental evaluation, proofs, (pseudo-)code, references, assumptions).

**Q2-4 Reproducibility:**

4: Excellent: key resources (e.g. proofs, code, data) are available and key details (e.g. proof sketches, experimental setup) are comprehensively described for competent researchers to confidently and easily reproduce the main results.

**Q3 Main Strengths:**

* The paper introduced a novel approach utilizing the DRSL framework for tackling density estimation.
* The training technique proposed is notably efficient. Specifically, the inner optimization is exact and requires time linear to the sample size. The outer optimization process is straightforward and uncomplicated.
* Experimental results highlight the effectiveness of the DRSL framework, showcasing improved test log likelihood on examples shifted from the training distribution compared to the estimates derived from MLE.

**Q4 Main Weakness:**

I found the evaluation is somewhat weak in the paper. In particular, the paper only compares the DRSL estimate with the MLE estimate. It missed the comparison with other robust density estimation work with different flavors.

For example, [Peddi 2022](https://proceedings.mlr.press/v180/peddi22a.html) proposed to learn robust TPMs by assuming a particular structure of distribution shift, i.e XOR on the discrete input examples. How does the DRSL estimate behave under those situations? Can the DRSL estimates robustly estimating the test log likelihood without assuming the noise structure?

**Q5 Detailed Comments To The Authors:**

The title suggests the algorithm is designed for TPMs. Is there any limitation of applying the algorithm on general density estimator that is trained with the decomposable loss on the training example?

Could author provide more information about the convergence behavior of the algorithm? For example, are there example of the dataset where the training won't be stable? How does a typical learning curve look like for the experiments after each iteration of the inner and outer optimization?

In the experiment, author removes every pair of attributes whose Pearson correlation coefficient is greater than 0.98. Can author provides more context on the decision?

On table 2 and 3, average LL from all the experiments are reported. I found average might be a misleading metrics here as the winning from a single benchmark might dominate, especially the test log likelihood are not comparable across different benchmarks. Maybe the # wins # losses could be a better metrics?

**Q9 Complying With Reviewing Instructions:**

Yes

---

> ### Author Rebuttal · Authors · 2024-04-08
>
> Thank you for your time reviewing our paper!
>
> Peddi's work primarily focuses on the discrete domain and can be categorized as Adversarially Robust Maximum Likelihood Estimation (MLE). It optimizes against the worst neighbor defined within certain structured constraints. In contrast, our research focuses on distributional shift in continuous domains, where no specific structure of corruption is assumed here.
>
> While our methods can indeed be applied to the discrete case and other general probabilistic models that are not tractable, our primary emphasis in this paper lies on tractable models. This choice is motivated by the fact that the learning step can usually be addressed more efficiently in such models.
>
> While we have proposed an exact and efficient algorithm for the adversarial step, we recognize that the overall iterations between the learning and adversarial steps can sometimes diverge. Minimax optimization remains an open problem, and we adopt strategies similar to those employed in training Generative Adversarial Networks (GANs).
>
> Regarding the procedure for removing highly correlated variables ($> 0.98$) in the data, we would like to clarify that our approach aligns with preprocessing procedures used in the existing literature, the references are included in the paper.
>
> We appreciate your insight and will add an additional table to provide a more comprehensive presentation of the results in our paper.

---

### Official Review · Reviewer_xRsH · 2024-03-20

**Q2-1 Originality-Novelty:** 3
**Q2-2 Correctness-Technical Quality:** 3
**Q2-5 Clarity Of Writing:** 3

**Q1 Summary And Contributions:**

This paper considers the problem of learning a distributionally robust tractable probabilistic model in continuous domains.
The contribution leverages Distributionally Robust Supervised Learning (DRSL), interleaving an adversarial step and a learning step (weighted MLE). The authors show that the adversarial optimization step can be solved globally very efficiently when using the KL divergence as a metric among distributions. The approach is empirically compared with the non-robust variant on multiple UCI datasets and MNIST embeddings.

**Q2-3 Extent To Which Claims Are Supported By Evidence:**

2: Fair: the main claims are somewhat supported by evidence (but the experimental evaluation may be weak, or does not match entirely with the claims, important baselines may be missing, proofs contain important ideas but lack rigor, algorithmic details are only discussed superficially, references are imprecise, assumptions are not sufficiently motivated or explicated, etc.).

**Q2-4 Reproducibility:**

3: Good: key resources (e.g. proofs, code, data) are available and key details (e.g. proofs, experimental setup) are sufficiently well-described for competent researchers to confidently reproduce the main results.

**Q3 Main Strengths:**

In my opinion, the paper is well written and easily accessible to a non-expert audience.
The problem considered is relevant in many application domains and the proposed approach seems sound to me, albeit I haven't carefully checked the proofs. The experimental section seems to confirm the validity of the approach and is sufficiently detailed to reproduce the results.

**Q4 Main Weakness:**

I think that the evaluation could be improved (see Q5).
The limitations of the proposed approach are not explicitly discussed.

**Q5 Detailed Comments To The Authors:**

In the experimental section, I wish that the authors reported mean and standard deviation for their results.
Currently, it is not clear how robust (to different initial conditions) the performance gain is. Is it prohibitive to train the models multiple times with different seeds?
Also, I wish that the section explicitly listed and addressed research questions to further improve readability. For instance, it is not completely clear to me what question are the Worst/Avg neighborhood metrics addressing.
Given that distributional robustness is the motivation for this work, I wish that the paper reported results over different values for $\delta$.
The choice of parameters for Gaussian and Jitter noise (Sec. 4.2) seems quite arbitrary. I also don't understand why clipping the noise in the former case, which makes it very similar to the latter.

**Q9 Complying With Reviewing Instructions:**

Yes

---

> ### Author Rebuttal · Authors · 2024-04-08
>
> We sincerely appreciate the time you've taken to review our paper.
>
> The Worst/Avg neighborhood metrics serves primarily as a means to explore whether the learning distribution exhibits smoothing characteristics, rather than simply concentrating a large amount of probability mass near the data points.
>
> In terms of the choice of delta, we followed the values used in the existing literature, the detailed references can be found in the paper. Furthermore, we conducted experiments across multiple datasets and models to provide empirical evidence supporting the efficacy of our approach.

---

### Official Review · Reviewer_PQpb · 2024-03-23

**Q2-1 Originality-Novelty:** 2
**Q2-2 Correctness-Technical Quality:** 3
**Q2-5 Clarity Of Writing:** 3

**Q1 Summary And Contributions:**

The authors present an approach building upon distributionally robust supervised learning for tractable probabilistic models. The approach presented by the authors imitates a coordinate ascent style algorithm, where for a set of model parameters, the authors perform an adversarial step, and compute the weight vectors, which are essentially a probability ratio of a data point in the training set belonging to the worst adversarial distribution over the in-distribution. These weight vectors are then used to optimize the model parameters in the outer loop. The authors present empirical results on MNIST + UCI datasets and show results on log likelihood improvements on noisy versions of the distribution test sets.

**Q2-3 Extent To Which Claims Are Supported By Evidence:**

2: Fair: the main claims are somewhat supported by evidence (but the experimental evaluation may be weak, or does not match entirely with the claims, important baselines may be missing, proofs contain important ideas but lack rigor, algorithmic details are only discussed superficially, references are imprecise, assumptions are not sufficiently motivated or explicated, etc.).

**Q2-4 Reproducibility:**

4: Excellent: key resources (e.g. proofs, code, data) are available and key details (e.g. proof sketches, experimental setup) are comprehensively described for competent researchers to confidently and easily reproduce the main results.

**Q3 Main Strengths:**

- On a high level, the paper presents an approach for building more robust models that generalize better under noisy/adversarial test inputs. This is a useful direction for the UAI community.

- The paper is well written, and the methodology seems straightforward from an implementation perspective. I also commend the authors for providing the code for their work, which is essential for reproducibility.

**Q4 Main Weakness:**

- I am concerned about the novelty of this work. This seems like a straightforward extension of DRSL, and very similar to adversarial learning approaches. The main novelty here seems to be converting this to a Lagrangian dual problem.

- Overall the scope and impact of this work seems low, given that the authors have only experimented on tractable probabilistic models, without giving any idea on how this might be extended to larger models.

- The experimentation doesn't include any baselines. I'd encourage the authors to look at other adversarial learning approaches, or Bayesian approaches and provide some baselines for this work.

**Q5 Detailed Comments To The Authors:**

- Can the authors highlight motivation behind their approach of only focusing on TPMs? Is there anything specific in their approach that won't extend to other more complex non linear models such neural networks?

- In section 1, the authors mention "ARM optimizes against the worst-case scenarios among neighboring data observations within a certain distance, treating each neighboring point with equal impor- tance, a practice that might not align with real-world scenarios." Can you explain why?

- You should include Bayesian baselines and other adversarial learning approaches in the experimental section. See refs [1, 2] for examples on each of them.

- Why is $\log{r_i}^{r_i}$ more stable numerically than ${r_i}\log{r_i}$?

- In section 3.3, it says that "The adversarial step is usually significantly faster (typically around 1-5 seconds), compared to the learning step that often takes minutes or even hours". Why such a high time for learning step? I though the adversarial and learning steps are interleaved? Is it because we perform more learning updates per adversarial updates? Can you please provide a breakdown on this?

- For the Gaussian Mixture Model, I am assuming that both the EM steps happen within the learning step. Please correct me if I am wrong.

References

[1] Vadera, Meet, Jinyang Li, Adam Cobb, Brian Jalaian, Tarek Abdelzaher, and Benjamin Marlin. "URSABench: A system for comprehensive benchmarking of Bayesian deep neural network models and inference methods." Proceedings of Machine Learning and Systems 4 (2022): 217-237.

**Q9 Complying With Reviewing Instructions:**

Yes

---

> ### Author Rebuttal · Authors · 2024-04-08
>
> Thank you for your time reviewing our paper!
>
> Regarding the novelty and contribution of our paper, we tackle the problem of learning distributionally robust probabilistic models. The study of tractable models is well-established, and the robust learning of these probabilistic models is of great interest to the research communities of tractable probabilistic models. To the best of our knowledge, this is the first paper that explored the application of DRO for learning generative tractable probabilistic models, instead of discriminative models. A key observation is that the negative log-likelihood can serve as a surrogate loss function. In addition, another contribution lies in devising a much more efficient algorithm (can be 10x or even 100x faster compared to general gradient descent based solvers) tailored for our KL-DRO problem.
>
> The effectiveness of our algorithm and method for learning generative tractable probabilistic models is empirically demonstrated through comprehensive experiments conducted on both adversarial data and original uncorrupted data. These experiments validate the robustness and applicability of our approach across different scenarios, further substantiating the significance of our contributions.
>
>
> For your detailed comments:
>
> While our methods can indeed be applied to the discrete case and other general probabilistic models that are not tractable, our primary emphasis in this paper lies on tractable models. This choice is motivated by the fact that the learning step can usually be addressed more efficiently in such models.
>
> In response to concerns about treating all neighbors equally, we present a simple digit image example to illustrate why this may not be optimal. Consider defining neighbors of a binary image as those images with at most k=5 different pixels. In such a scenario, an image neighbor with pixel differences occurring within the actual digit holds more significance than an image neighbor with differing pixels located at the image corners.
>
> The adversarial step is way faster because the design of our algorithm. It is not based on general gradient descent (commonly used in learning step) that can be slow in the case of large data and expressive neural density functions.

---

### Official Review · Reviewer_kQr7 · 2024-03-24

**Q2-1 Originality-Novelty:** 3
**Q2-2 Correctness-Technical Quality:** 3
**Q2-5 Clarity Of Writing:** 4

**Q1 Summary And Contributions:**

This paper studies distributionally robust learning of continuous tractable probabilistic models. The authors employ the distributionally robust supervised learning (DRSL) framework with KL-divergence as the distributional distance metric and negative log-likelihood as the loss to formulate the optimization problem. They show that the dual of this problem can be solved as a minimax game with an efficient, exact algorithm for the adversarial step (inner optimization) and the learning step (outer optimization) reducing to standard MLE on weighted datasets.

**Q2-3 Extent To Which Claims Are Supported By Evidence:**

3: Good: the main claims are supported by convincing evidence (in the form of adequate experimental evaluation, proofs, (pseudo-)code, references, assumptions).

**Q2-4 Reproducibility:**

3: Good: key resources (e.g. proofs, code, data) are available and key details (e.g. proofs, experimental setup) are sufficiently well-described for competent researchers to confidently reproduce the main results.

**Q3 Main Strengths:**

The proposed approach is derived from a fairly straightforward application of DRSL to negative log-likelihood loss which is intuitive, and the resulting algorithm is elegant with an exact solution for the inner optimization and standard MLE for outer optimization.

The paper is overall well-written and easy to follow. Proofs and derivations in the main body are all correct as far as I can tell.

**Q4 Main Weakness:**

If there are any assumptions about the probability density (which I think there should be), this was not clear from the paper. Does the density function have to be continuous? Why is the approach applicable to continuous domains but not discrete?

The paper is missing a discussion on related and prior work.

In Tables 2 and 3, comparing the average LL scores over all datasets can be misleading since the LL is unbounded for densities, as the authors also point out in the paper. I would suggest looking into how many datasets DRSL outperformed MLE. For example, on three out of 9 datasets for MixMG and one dataset for NN-GBN, DRSL performed worse than MLE across all test cases.

**Q5 Detailed Comments To The Authors:**

While the authors introduced the work as distributional robust learning for tractable probabilistic models, the method seems to be applicable to any probabilistic model with explicit likelihood. In fact, none of the examples of TPMs mentioned in the introduction were used in the evaluation. I hope the authors can clarify.

Does the density function have to be continuous? Why is the approach applicable to continuous domains but not discrete?

Some discussion (theoretical or empirical) of the convergence of the minimax approach would be interesting. How long does DRSL take to converge compared to MLE? Could the adversarial and learning steps keep oscillating?

KLD(Q,P) seemed like an odd choice compared to KLD(P,Q) which is a more common objective for learning Q. Is this standard in DRSL formulation?

I did not quite follow how neighboring data points were sampled for Worst NB and Average NB. Are these based on Gaussian noise and jittering inputs described previously?

For MNIST experiments: does distributional robustness in the latent feature space translate to robustness in the image input space? On a related note, Worst NB and Average NB can still be done by sampling neighbors in the image space, as I think you ultimately want robustness in the input space.

**Q9 Complying With Reviewing Instructions:**

Yes

---

> ### Author Rebuttal · Authors · 2024-04-08
>
> We sincerely appreciate the time you have taken to review our paper.
>
> In response to the concern regarding the assumption of probability density,  our approach aligns with the assumptions commonly made in the Maximum Likelihood Estimation (MLE) framework. This is because the learning step essentially entails MLE on weighted data.
>
> We are interested in the continuous domain because it is more challenging and the robust learning for probabilistic models in continuous spaces is under explored.
>
> Regarding the comparison of log-likelihoods, we are solely comparing the performance of standard MLE against DRO on the same type of model. Therefore, log-likelihood serves as a valid metric in such cases.
>
> Addressing your detailed comments, while our methods can indeed be applied to the discrete case and other general probabilistic models that are not tractable, our primary emphasis in this paper lies on tractable models. This choice is motivated by the fact that the learning step can usually be addressed more efficiently in such models. However, we are keen on exploring the application of our method on other models, such as SPNs, in future research endeavors.
>
> Regarding the DRSL formulation, it is from the existing literature. While we have proposed an exact and efficient algorithm for the adversarial step, we recognize that the overall iterations between the learning and adversarial steps can sometimes diverge. Minimax optimization remains an open problem, and we adopt strategies similar to those employed in training Generative Adversarial Networks (GANs).
>
> The neighboring points are uniformly sampled in the epsilon hyperball around the actual data points.
>
> We agree that investigating whether distributional robustness in the latent feature space translates to robustness in the image input space can be interesting. However, as the encoder for image (VAE in our case) already provides a noticable amount of robustness, it is difficult to know how much robustness is contributed by DRSL form the final results.

---

### Official Review · Reviewer_4SFJ · 2024-03-27

**Q2-1 Originality-Novelty:** 1
**Q2-2 Correctness-Technical Quality:** 2
**Q2-5 Clarity Of Writing:** 2

**Q1 Summary And Contributions:**

This paper proposes a distributionally robust optimization algorithm for probabilistic models. The proposed method is the same as KL-DRO, and experimental results demonstrate its effectiveness.

**Q2-3 Extent To Which Claims Are Supported By Evidence:**

2: Fair: the main claims are somewhat supported by evidence (but the experimental evaluation may be weak, or does not match entirely with the claims, important baselines may be missing, proofs contain important ideas but lack rigor, algorithmic details are only discussed superficially, references are imprecise, assumptions are not sufficiently motivated or explicated, etc.).

**Q2-4 Reproducibility:**

2: Fair: key resources (e.g. proofs, code, data) are unavailable but key details (e.g. proof sketches, experimental setup) are sufficiently well-described for an expert to confidently reproduce the main results.

**Q3 Main Strengths:**

- The authors propose to use KL-DRO for continuous TPMs.
- Extensive results are provided.

**Q4 Main Weakness:**

- Novelty: I think the proposed DRO formulation is **exactly the same as KL-DRO**, which is a well-known and well-developed algorithm. Therefore, I think the technical novelty of this paper is quite poor. I would recommend the authors to do a thorough literature review of DRO, for example, [1,2,3]. Furthermore, the incorporation of continuous TPMs does not lead to a new problem formulation.
- Baselines: There lack many baselines in the experiments, such as other DRO methods and some robust learning methods.




[1] Rahimian, H., & Mehrotra, S. (2019). Distributionally robust optimization: A review. arXiv preprint arXiv:1908.05659.
[2] Duchi, J. C., & Namkoong, H. (2021). Learning models with uniform performance via distributionally robust optimization. The Annals of Statistics, 49(3), 1378-1406.
[3] The DRO package: https://github.com/namkoong-lab/dro

**Q5 Detailed Comments To The Authors:**

Please refer to the weaknesses.

**Q9 Complying With Reviewing Instructions:**

Yes

---

> ### Author Rebuttal · Authors · 2024-04-08
>
> We appreciate your review of our paper and your valuable expertise in the field of Distributionally Robust Optimization (DRO) and probabilistic models. Note that we have provided the full codes in the supplementary material, and the formal proofs are available in the appendix. The proofs show why our algorithm consistently yields optimal solutions within any given error threshold and converges effectively.
>
> Regarding the novelty and contribution of our paper, we tackle the problem of learning distributionally robust probabilistic models. The study of tractable models is well-established, and the robust learning of these probabilistic models is of great interest to the research communities of tractable probabilistic models. To the best of our knowledge, this is the first paper that explored the application of DRO for learning generative tractable probabilistic models, instead of discriminative models. A key observation is that the negative log-likelihood can serve as a surrogate loss function. In addition, another contribution lies in devising a much more efficient algorithm (can be 10x or even 100x faster compared to general gradient descent based solvers) tailored for our KL-DRO problem.
>
> The effectiveness of our algorithm and method for learning generative tractable probabilistic models is empirically demonstrated through comprehensive experiments conducted on both adversarial data and the original uncorrupted data. These experiments validate the robustness and applicability of our approach across different scenarios, further substantiating the significance of our contributions.

---

### Meta-Review · Area_Chair_jjYM · 2024-04-20

The paper introduces a distributionally robust method to train tractable probabilistic models such as probabilistic circuits. In particular, the method benefits from exact likelihoods and the ability to learn from weighted data.

The paper is technically sound and expands the range of techniques to train tractable models on imprecise data. The experiments show robustness of the method under various data corruptions. The novel aspect of the paper is its application of DRO to density estimation. The particular restriction to tractable models is not entirely clear, and it is questionable whether the paper will have a larger impact.

The reviewers who contributed in the discussion were on the positive side, recommending accept, albeit still on the borderline side.